# Stability Constraints on Practical Implementation of Parity-Time-Symmetric Electromagnetic Systems

Josip Lončar *[ID], Josip Vuković [ID], Igor Krois and Silvio Hrabar

Faculty of Electrical Engineering and Computing, University of Zagreb, Unska 3, 10000 Zagreb, Croatia; josip.vukovic@fer.hr (J.V.); igor.krois@fer.hr (I.K.); silvio.hrabar@fer.hr (S.H.)
* Correspondence: josip.loncar@fer.hr

**Abstract:** Recently, several applications leveraging unconventional manipulation of electromagnetic radiation based on parity-time symmetry have been proposed in the literature. Typical examples include systems with unidirectional invisibility and asymmetric refraction. Such applications assume an inherent system stability and no occurrence of unbounded signal growth or unwanted self-oscillations. Here, a general instability issue of parity-time-symmetric systems is investigated, with particular emphasis on a recently proposed system based on resistive metasurfaces. Explicit closed-form stability criterion is derived, crosschecked and verified by both time-domain transient simulations and the measurements on an experimental demonstrator operating in a lower radiofrequency range. Results of this study lead to the conclusion that any parity-time-symmetric system is necessarily marginally stable. Finally, it is shown that such a marginally stable system may easily become unstable if not designed carefully.

**Keywords:** PT symmetry; metasurfaces; stability analysis; negative-impedance converter

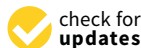



## 1. Introduction

In the last few years, a significant research effort has been devoted to the concept of parity-time (PT) symmetry. Originally, this principle comes from quantum mechanics. It states that non-Hermitian Hamiltonian $H$, which commutes with the parity-time (PT) operator, can have real-energy eigenvalues [1,2]. Comparing the Schrödinger equation and the scalar Helmholtz equation [3], i.e., comparing the linear operators in quantum mechanics and the matrix description of electrical networks [4,5], the principles of PT symmetry have recently been extended to electronics [4], electromagnetics [3,6], optics [7] and acoustics [8,9]. In their simplest one-dimensional form, the electromagnetic PT-symmetric systems are based on a combination of passive and active slabs with the refractive indices that form a complex conjugate pair [7]. As an active system (a system with its own source of energy), a PT-symmetric system can operate in two fundamentally different modes: Unstable mode and stable mode. In unstable mode, a PT-symmetric system converts the energy from its power supply into self-oscillations. A practical application of unstable mode is a PT-symmetric laser [10]. On the other hand, in stable mode, there is no self-oscillation. A stable system, excited with a bounded input signal, always responds with a bounded output signal (so-called Bounded Input Bounded Output or BIBO stability). There are many proposed applications of PT–symmetric systems that assume BIBO stability, such as unidirectional invisibility, loss-induced transparency, asymmetric refraction, perfect absorption, and non-trivial anisotropic transmission resonance [7,8], just to mention a few. In contrast to volumetric structures, some of the aforementioned effects can also be achieved using a simple pair of metasurfaces represented with positive and negative surface resistance [3,11]. The metasurfaces with positive surface resistance attenuate the power of an incident electromagnetic wave. The structure of such metasurfaces is based on passive, lossy materials [12]. On the other hand, the metasurfaces with negative surface resistance amplify an incident wave, and thus require an internal power source and active

circuitry [13–15]. Ensuring BIBO stability is a must in the applications listed above. As clearly noted in [8], the inherent constraints of PT-symmetric systems dictated by causality and stability cannot be overlooked in practical implementation. There has been a very recent attempt to achieve some of the counter-intuitive phenomena that arise from PT-symmetry in fully passive, inherently stable systems [16]. In such a way, the instability issue would be avoided. Unfortunately, very few studies reported in the literature cope with the stability of active PT-symmetric systems in general. Most of them are associated with highly specific PT-symmetric systems, use theoretical methods of quantum mechanics and lack experimental verification [17–21]. Therefore, we report the stability analysis of PT-symmetric systems in general. Furthermore, the stability analysis of the recently proposed metasurface-based PT-symmetric system [3] is detailed. We report a derivation of the stability criterion and an analysis of its natural responses, along with the numerical and experimental verification of the presented theoretical approach.

## 2. Stability of Parity-Time-Symmetric Systems

PT-symmetric systems are often referred to as systems with a balanced distribution of gain and loss [8]. While this characterization is not strictly mathematical, it conveys information about the stability. Indeed, if the gain and loss within a system are perfectly balanced, the system is marginally stable. If the loss prevails over the gain, the energy within the system dissipates over time, making the system stable. Similarly, if the gain prevails over loss, the energy within the system unboundedly accumulates, which is by definition the manifestation of instability. This leads to the conclusion that PT-symmetric systems are necessarily marginally stable. In a mathematical sense, the PT symmetry is a special type of space–time symmetry that describes the invariance of a physical system upon the combined action of two operators: The parity operator P, which takes the inversion of space coordinates ($z \rightarrow -z$), and the time reversal operator T, which reverses the sign of the time variable ($t \rightarrow -t$) [8]. From the circuit-theory point of view, the action of the operator P is to the mirror spatial layout, and the one of the operator T is to switch gain and loss [8]. Thus, upon the action of the operator T, the real part of all network impedances changes the sign. Notice that the operator P does not affect the stability properties of a system to which it is applied. However, this is not the case with the operator T. Let us assume that the operator T is applied to a stable system with a natural response that decays with time. Upon the action of the operator T, the sign of a time variable is inverted, making its natural response grow unboundedly. Thus, the operator T makes a stable system unstable, and vice versa. If a system changes its stability properties after applying the operator T, it cannot remain unchanged upon the action of both operators, P and T (i.e., the system is not invariant to the combined action of the two operators). Therefore, such a system is not PT-symmetric. Only if a system does not change its stability properties upon the action of the operator T can it be PT-symmetric. This is possible only if the natural response of a system does not decay or grow with time, which is a property of marginally stable systems. This again leads to the conclusion that only marginally stable systems can be PT-symmetric. Thus, every PT-symmetric system is indeed marginally stable.

To verify this statement, we conducted a stability analysis of the recently proposed metasurface-based PT-symmetric system [3]. Its one-dimensional circuit model, shown in Figure 1, consists of a parallel combination of a positive and a negative resistor, mutually connected via a segment of ideal transmission line described by its length ($l$), characteristic impedance ($Z_0$) and phase velocity ($v_P$). Following the recently published stability analysis of distributed networks with negative elements [22–24], the locations of the poles $s$ of the network from Figure 1 can be found by solving the system equation:

$$\tanh\left(\frac{sl}{v_P}\right) = -Z_0 \frac{R_L + R_R}{R_L R_R + Z_0^2}. \tag{1}$$

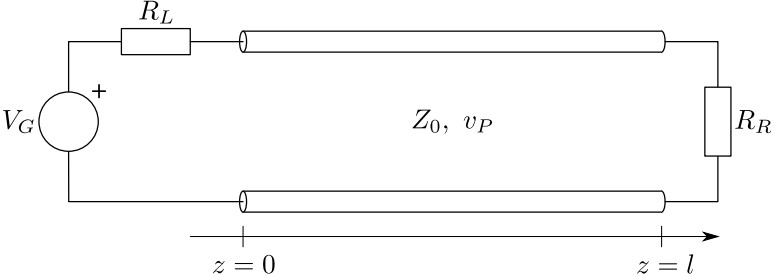

**Figure 1.** A generalized circuit model of the recently proposed metasurface-based PT-symmetric system [3].

Here, $s$ stands for complex frequency defined as $s = \sigma + j\omega$ ($e^{st}$ convention is used). In the most general case, the right-hand side of (1) contains complex impedances, thus, it is a function of complex frequency $s$. In such a general case, (1) is a transcendental equation that cannot be solved analytically [3]. Here, however, the right-hand side of (1) is not a function of complex frequency $s$, since $R_L$, $R_R$, and $Z_0$ are positive real constants. Thus, the closed-form expression for pole locations can be derived:

$$s = \frac{v_P}{2l}\left\{\ln|\Gamma_L\Gamma_R| + j[\text{Arg}(\Gamma_L\Gamma_R) + 2k\pi]\right\}. \tag{2}$$

Equation (2) reveals an infinite number of poles ($k \in \mathbb{Z}$) related to the periodic behavior of the transmission line. Here, $\Gamma_L, \Gamma_R \in \mathbb{R}$ represent reflection coefficients given by the well-known expressions [25]:

$$\Gamma_L = \frac{R_L - Z_0}{R_L + Z_0}, \tag{3a}$$

$$\Gamma_R = \frac{R_R - Z_0}{R_R + Z_0}. \tag{3b}$$

It is interesting that all the poles are aligned along the line parallel to the imaginary axis of the complex plane. It is well known that, for a stable operation, all poles of a system must lie in the left half-plane of complex plane ($\text{Re}\{s\} < 0$). This condition is satisfied if $\ln|\Gamma_L\Gamma_R| < 0$, which leads to the general stability criterion:

$$|\Gamma_L\Gamma_R| < 1. \tag{4}$$

Please notice that stability of the system does not depend on the length of transmission line $l$, but $R_L$, $R_R$ and $Z_0$ only. The stable and unstable combinations of $R_L$, $R_R$ and $Z_0$ can be determined from the graph in Figure 2. Shadowed areas represent the regions of stable operating points that satisfy stability criterion (4). Stable and unstable regions are separated by two stability margins given by (5). Those margins are derived from the condition for marginal stability ($\text{Re}\{s\} = 0 \rightarrow |\Gamma_L\Gamma_R| = 1$).

$$R_L = -R_R \tag{5a}$$

$$R_L R_R = -Z_0^2 \tag{5b}$$

If $R_L$ and $R_R$ are chosen from the first quadrant, both resistors are positive and dissipate injected energy upon each reflection. As a result, the response decays with time and the network is stable. If $R_L$ and $R_R$ are chosen from the third quadrant, both resistors are negative. Upon each reflection from negative resistors the amount of energy in the network increases. As a result, the response unboundedly grows with time, thus the network is unstable. However, when it comes to PT-symmetric systems, one is interested in a combination of positive and negative resistors (the second and the fourth quadrant). Recall that a network is PT symmetric only if it remains unchanged upon the combined action of the operators P and T. Therefore, the network from Figure 1 is PT symmetric only if the resistors $R_L$ and $R_R$ have the same absolute value and opposite signs. This condition

places the operating point of the network on the line defined by (5a), representing the stability margin. As a result, the analyzed PT-symmetric network is indeed marginally stable. In this case the resistors can be related to the characteristic impedance as $R_L = rZ_0$ and $R_R = -rZ_0$, $r \in \mathbb{R}$, being a proportionality constant. It can be easily shown that $|\Gamma_L \Gamma_R| = 1$ for any $r$.

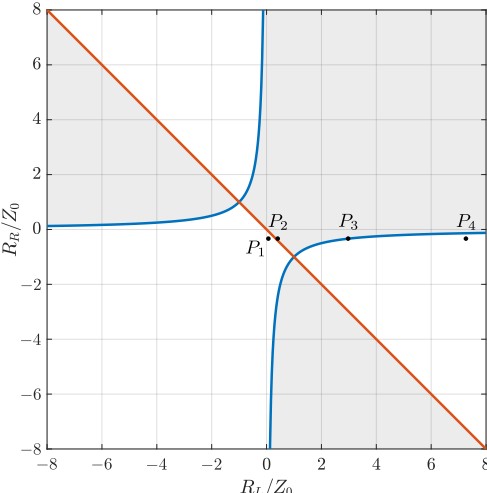

**Figure 2.** Regions of stable (shaded) and unstable (white) operation. In the shaded regions stability criterion (4) is satisfied.

While (4) clearly shows that only reflection coefficients $\Gamma_L$ and $\Gamma_R$ affect the stability of the network, both the length of the transmission line ($l$) and phase velocity ($v_P$) influence the pole locations in the complex plane. These parameters determine the type of natural response of the network. By analyzing the poles (2), it is possible to predict the rate of growth and repetition frequency of the natural response. The envelope of the response is defined by the exponential function:

$$v(t) = V_0 e^{\sigma t} \; \rightarrow \; v(t) = V_0 |\Gamma_L \Gamma_R|^{\frac{t}{2\tau}}. \tag{6}$$

Here, $\tau$ represents the transmission line delay defined as $\tau = l/v_p$. The real part of the poles $\sigma$ represents the rate of growth. As (6) indicates, the natural response decays with time only if $|\Gamma_L \Gamma_R| < 1$, which is consistent with the stability criterion (4).

The repetition frequency of natural response is defined by the smallest imaginary part of the poles different than zero. According to (2), if $\Gamma_L \Gamma_R > 0 \; \rightarrow \; \text{Arg}(\Gamma_L \Gamma_R) = 0$, a pole occurs at the real axis of the complex plane for $k = 0$. As a result, the natural response is an exponential-like direct current (DC) signal. In this case, the repetition frequency is defined for $k = 1$, leading to (7a). If $\Gamma_L \Gamma_R < 0 \; \rightarrow \; \text{Arg}(\Gamma_L \Gamma_R) = \pi$, there is no pole on the real axis. Thus, the response is purely oscillatory, with the repetition frequency defined by (7b), for $k = 0$. In both cases the repetition frequency depends on $\tau$. Notice that the repetition frequency of the exponential-like response (7a) is twice the repetition frequency of the oscillatory response (7b).

$$\omega_0 = \frac{v_P}{2l}[\text{Arg}(\Gamma_L \Gamma_R) + 2k\pi] \xrightarrow[\Gamma_L \Gamma_R > 0]{k=1} \omega_0 = \frac{1}{2\tau}[0 + 2\pi] \; \rightarrow \; f_0 = \frac{1}{2\tau} \tag{7a}$$

$$\omega_0 = \frac{v_P}{2l}[\text{Arg}(\Gamma_L \Gamma_R) + 2k\pi] \xrightarrow[\Gamma_L \Gamma_R < 0]{k=0} \omega_0 = \frac{1}{2\tau}[\pi + 0] \; \rightarrow \; f_0 = \frac{1}{4\tau} \tag{7b}$$

According to the analysis given above, it is possible to define four types of natural response: Stable exponential-like ($|\Gamma_L \Gamma_R| < 1$, $\Gamma_L \Gamma_R > 0$), unstable exponential-like

$(|\Gamma_L\Gamma_R| > 1, \Gamma_L\Gamma_R > 0)$, stable oscillatory $(|\Gamma_L\Gamma_R| < 1, \Gamma_L\Gamma_R < 0)$ and unstable oscillatory $(|\Gamma_L\Gamma_R| > 1, \Gamma_L\Gamma_R < 0)$ response.

## 3. Numerical and Experimental Verification

The presented theoretical analysis was verified both by time-domain transient simulations using a commercial circuit-theory solver Keysight ADS™ and by measurements on a designed and constructed low-frequency experimental demonstrator.

### 3.1. Simulations of the Equivalent Circuit Model

The simulation model, based on the circuit from Figure 1, comprised a positive resistor $R_L$ that was connected to the negative resistor $R_R = -25\,\Omega$ via a segment of ideal transmission line. Clearly, it is a direct one-dimensional radio freqency analog of the active metasurface system proposed in [3]. The value of the positive resistor $R_L$ was varied in simulations. Positive $R_L$ and negative $R_R$ placed the operating point of the network in the fourth quadrant of the graph in Figure 2. The network was excited by a single voltage pulse and the voltage across the negative resistor $R_R$ was monitored. The waveforms obtained by time-domain simulations, representing the four types of natural response, are shown by solid red curves in Figure 3. The envelope and repetition frequency of all signals were determined using Equations (6) and (7), and compared with the simulated results in Figure 3. It was found that the simulated results are in absolute agreement with all theoretical predictions.

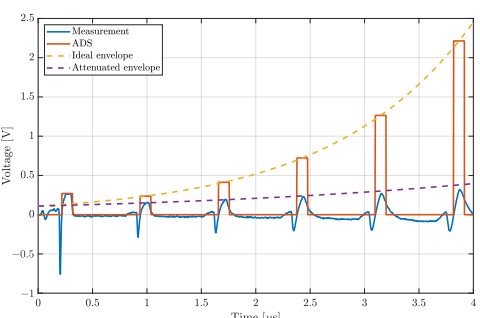

(**a**) Unstable exponential-like response (point $P_1$ in Figure 2, $R_L = 5\,\Omega$, $R_R = -25\,\Omega$).

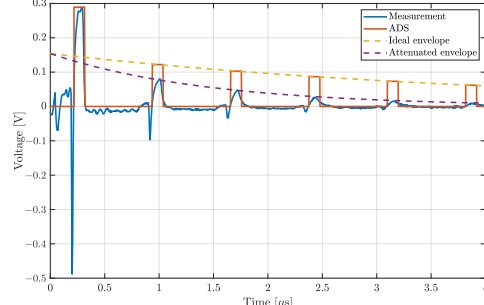

(**b**) Stable exponential-like response (point $P_2$ in Figure 2, $R_L = 30.5\,\Omega$, $R_R = -25\,\Omega$).

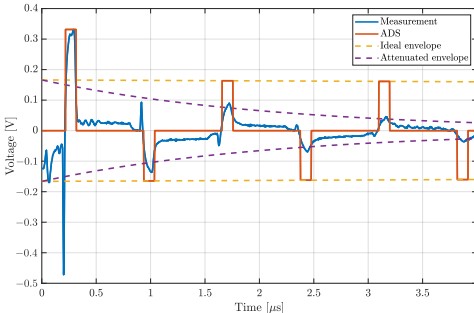

(**c**) Stable oscillatory response (point $P_3$ in Figure 2, $R_L = 223\,\Omega$, $R_R = -25\,\Omega$).

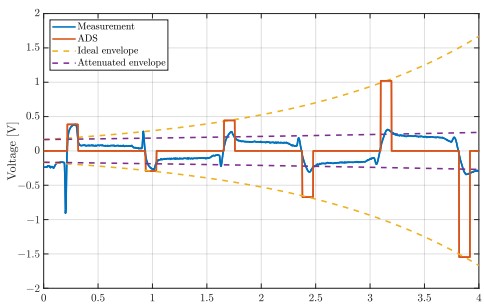

(**d**) Unstable oscillatory response (point $P_4$ in Figure 2, $R_L = 545\,\Omega$, $R_R = -25\,\Omega$).

**Figure 3.** Four types of natural response. Solid and dashed red curves represent response signals and their envelopes, analytically calculated using (6) and presuming ideal lossless transmission line. Solid blue curve represents measurement data while the dashed purple curve corresponds to envelope calculated by improved analytical model that takes losses into account. Simulation parameters: excitation pulse with magnitude of 1 V, the pulse width of 100 ns, rise time of 2 ns, fall time of 2 ns, $R_R = -25\,\Omega$, $Z_0 = 75\,\Omega$ and $\tau = 360$ ns.

### 3.2. Transmission-Line-Based Experimental Demonstrator

In the next step, the correctness of the proposed stability criterion was crosschecked and verified experimentally. Due to simplicity and repeatability, a negative resistor based on a Negative Impedance Converter (NIC), similar to those reported in [4,26], was designed and constructed. Here, in particular, a voltage-inverting NIC (VNIC) was used. The VNIC, shown in Figure 4a, uses a single operational amplifier (OPAMP) to "negate" the feedback resistor $R_F$, thus generating negative resistance at the input terminals, in the same manner as the current-inverting NIC (CNIC) detailed in [26]. Its input impedance is given with the following expression:

$$Z_{IN} = R_R = \frac{V_{IN}}{I_{IN}} = \frac{R_F}{1 - G_0} = -\frac{R_1}{R_2} R_F. \tag{8}$$

Here, $G_0$ represents the closed-loop gain of the OPAMP defined by the resistors $R_1$ and $R_2$ ($G_0 = 1 + R_2/R_1$). For $G_0 > 1$, the current passing through resistors $R_1$ and $R_2$ creates the voltage across the resistor $R_1$ that opposes the referent input voltage $V_{IN}$ (in stable mode, electric potentials of the positive and negative input terminals are equal, i.e., $V_D = 0$, $V_D$ being a differential input voltage). This voltage inversion indeed creates a scaled "negated" version of the resistance $R_F$ at the input port. Following the analysis proposed in [26], one can obtain a stability circle and a stability region within the Smith chart for the given negative resistance, representing all the passive impedances, loading the input port of the VNIC, that ensure stability. In contrast to CNIC, if used in a system described by the characteristic impedance $Z_0$, VNIC-generated negative resistance must satisfy condition $|R_R| < Z_0$ to maintain stability. Thus, in the 75 Ω system, a −25 Ω negative resistance is chosen, which defines the shaded stability region (i.e., allows passive loading impedances) shown in Figure 4b.

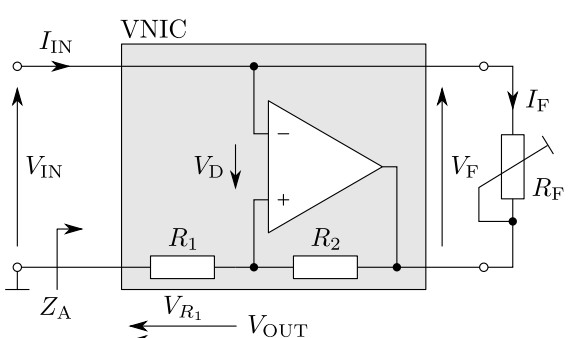 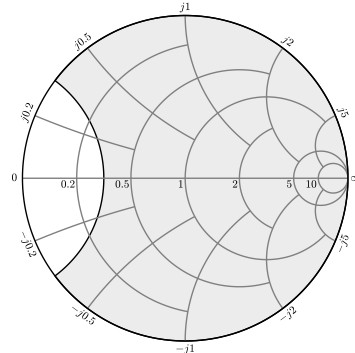

(**a**) A circuit model of VNIC-based negative resistance with denoted currents and voltages.

(**b**) Stable (shaded) region ($R_R = -25$ Ω, $Z_0 = 75$ Ω).

**Figure 4.** A circuit model of voltage-inverting negative impedance converter (VNIC)-based negative resistance and corresponding stability region within the Smith chart obtained by following the analysis similar to one proposed in [26] for generated negative resistance $R_R = -25$ Ω in a system with $Z_0 = 75$ Ω.

Before choosing an OPAMP that could be used in practical realization of the VNIC, its frequency characteristic (i.e., dispersion) must be taken into account. Equation (8) assumes an ideal OPAMP with constant gain and infinite bandwidth. However, the magnitude and phase of a realistic OPAMP vary with frequency. Thus, the closed-loop gain $G_0$ is not a constant, but a function of frequency. As frequency approaches infinity, the magnitude of the closed-loop gain inevitably drops to zero. This gain drop is predominantly dictated by the frequency of the first pole $f_P$ (dominant pole) of the closed-loop gain function. Such band-limited behavior of an OPAMP causes both the dispersion of the generated negative resistance and the occurrence of additional, unwanted input reactance (imaginary part of the input impedance). However, these effects are negligible if the

operating frequency is much lower than the frequency of the dominant pole $f_P$. Moreover, to minimize the influence of parasitic capacitance and inductance inevitably present in the radiofrequency regime, it is convenient to select the operating band of the demonstrator in a lower frequency range. Thus, the maximum operating frequency was set to 5 MHz. A high-speed dual OPAMP ADA4857-2 was selected for the realization of the VNIC circuit. With the dominant pole at approximately 100 MHz (for $G_0 = 2$), it meets the requirements and represents a suitable choice. Since the frequency of the dominant pole is much higher then the maximum operating frequency ($f_P > 10 f_{\max}$), it is expected that the input impedance generated by the VNIC should behave very closely to an ideal negative resistance. Based on the Equation (8), fixed resistors $R_1 = R_2 = 560\ \Omega$ and trimmer potentiometer $R_F$ with maximum resistance of 250 $\Omega$ were selected, and together with the OPAMP assembled with respect to the circuit shown in Figure 4a. All the components were soldered on a $2 \times 2\ \mathrm{cm}^2$ printed circuit board, equipped with an input SMA connector. Following the manufacturer's recommendations, the OPAMP was powered symmetrically using $\pm 5$ V DC voltage power supply.

The initial testing of the VNIC circuit was performed by measuring its input impedance using the Rohde & Schwarz ZNC3 Vector Network Analyzer (VNA). To ensure the stability of the VNIC during the measurement, an additional compensating resistor was used in series with the VNIC to form an equivalent positive resistance. The input impedance of the VNIC was extracted in post-processing, eliminating the influence of the series compensating resistor through a simple algebraic manipulation. This method revealed negative input resistance that can be adjusted from $-20\ \Omega$ to $-250\ \Omega$ with less than 10% error compared to the theoretical results, in the frequency range from 100 kHz to 5 MHz. In addition, the imaginary part of the input impedance was found to be negligible. Thus, the prototyped VNIC-based negative resistor shows satisfactory behavior in the frequency range of interest, as shown in Figure 5.

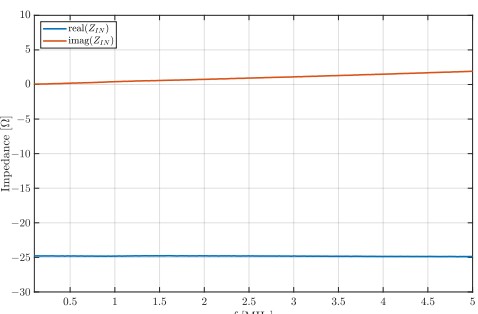

(**a**) Impedance $Z_{IN}$ (from 0.1 to 5 MHz).

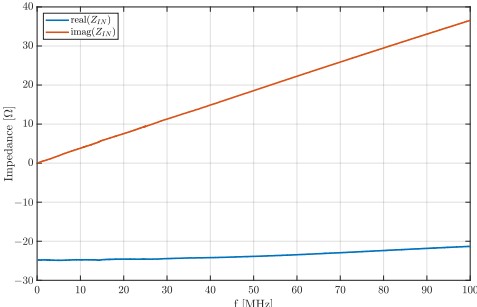

(**b**) Impedance $Z_{IN}$ (from 0.1 to 100 MHz).

**Figure 5.** Measurement of the input impedance generated by the VNIC, with the nominal input resistance $R_R = -25\ \Omega$, based on the circuit from Figure 4a obtained using the Rohde & Schwarz ZNC3 Vector Network Analyzer (VNA).

A standard trimmer potentiometer was used as a variable positive resistor $R_L$ with a maximum resistance of 600 $\Omega$. Both the variable positive resistor and the VNIC-based variable negative resistor $R_R$ were connected to the opposite ends of a 100 m long RG59B/U coaxial cable ($Z_0 = 75\ \Omega$, $\tau = 360$ ns). In this way, the normalized resistances $R_L/Z_0$ and $R_R/Z_0$ could be adjusted within a range from 0 to 8, and from 0 to $-3.3$, respectively. These values belong to the fourth quadrant from Figure 2. Due to the symmetry of the second and fourth quadrant, it was sufficient to reduce the investigation to the fourth quadrant only.

The stability was examined by measuring the voltage across the negative resistors using the LeCroy LT374L digital oscilloscope. The measured voltage was buffered using an additional OPAMP available within the ADA4857-2 integrated circuit in a high-impedance voltage follower configuration. The system response was measured for four different points of the fourth quadrant shown in Figure 2, corresponding to the four char-

acteristic types of natural response. Those four points represent the pairs of normalized impedances $R_L/Z_0$ and $R_R/Z_0$ set by adjusting trimmer potentiometers: $P_1(0.07, -0.33)$, $P_2(0.41, -0.33)$, $P_3(2.97, -0.33)$ and $P_4(7.27, -0.33)$. In order to successfully capture the transient response of the system that is present only a short period of time after the excitation, an additional synchronization mechanism is needed. Thus, an external Hewlett–Packard 222A pulse generator was used. It generates the "enable" signal fed to the Power Down (PD) pin of ADA4857-2. This signal "wakes up" the VNIC circuit while the pulse generator simultaneously triggers the time base of the oscilloscope. In this way, it is possible to capture an arbitrary part of the transient event. The complete experimental setup is detailed in Figure 6.

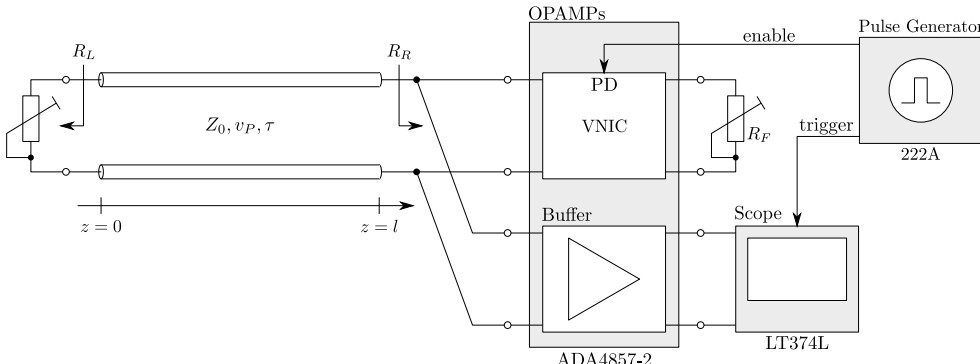

**Figure 6.** Complete measurement setup used for verification of the stability theory applied to PT-symmetric systems proposed in [3].

### 3.3. Simulation and Measurement Results

Measured signals (solid blue curves) are compared with the theory and simulations in Figure 3. At first, it can be seen that the waveform types are equivalent to those predicted by both theory and simulations: Unstable exponential-line (Figure 3a), stable exponential-line (Figure 3b), stable oscillatory (Figure 3c) and unstable oscillatory (Figure 3d). However, the measured waveforms evidently differ from the simulations. The reason is that the ideal rectangular pulse used as an excitation in the simulations is difficult to reproduce in experiments due to the parasitics and non-ideal behavior of the active circuitry. These imperfections, such as the band-limited and non-linear operation, also cause pulse reshaping upon each reflection. Another difference is in the amplitude of the response, which is noticeably lower than predicted. This discrepancy comes from the inevitable losses of the coaxial cable, which were not taken into account in the theoretical model. Based on the analysis given in Section 2, the losses within the coaxial cable contribute to the stability of the system, dissipating the power injected by the negative resistor in addition to the assumed dissipation of the positive resistor. Thus, the measured waveforms experience slower growth than the related simulated ones. The theoretical model was extended to account for the attenuation of the 0.02 dB/m characteristic for the RG59B/U coaxial cable. The envelopes obtained by the improved analytical model are shown by the dashed purple curves in Figure 3. It can be seen that the corrected envelopes closely follow the peaks of measured pulses, which validates our theoretical and numerical results.

### 4. Discussion on Limitations

One should be aware of yet another very important limitation, which lies in the implementation of the negative resistor circuitry. In general, a negative resistor is simply an electric element that relates the voltage $V$ across it to the current $I$ through it with the negative proportionality constant $R < 0$. Thus, it can be modeled using Ohm's law $V = RI$ extended to negative $R$. While such an element does not exist in nature, a circuitry that mimics its behavior can be designed using an NIC, as explained above. It is the designer's choice whether to use a voltage-inverting VNIC or a current-inverting CNIC. These two types of NICs show very different stability properties. Sometimes, they are classified as

open-circuit-stable (OCS) and close-circuit-stable (SCS), respectively. This classification, however, is uninformative concerning whether a circuit is stable when terminated with any other type of impedance [27,28]. A glimpse of a new promising approach to stability analysis of NIC-based devices, based on relating the positive and the negative feedback loop of an NIC, is given in [26]. Unfortunately, the stability properties of NICs are not yet fully understood by the scientific community. As a result, the circuitry that mimics the behavior of a negative resistor may show different stability properties that arise only from the implementation choice, which may introduce inconsistency in the stability analysis proposed here. To overcome this challenge, we decided to use a VNIC-based negative resistor with the fixed resistance $R_R = -25 \, \Omega$ that satisfies the condition $|R_R| < Z_0$, as stated in Section 3.2. Future research efforts will be devoted to the development of a comprehensive approach to stability analysis of NIC-based devices that avoids the aforementioned limitation.

After the criterion (4) is verified both numerically and experimentally, having in mind the limitation explained above, it can be safely used to accurately predict the stability properties of the PT-symmetric system proposed in [3]. The framed circuit shown in Figure 7 represents the equivalent circuit model of the analyzed system. Recall that for any $r \in \mathbb{R}$, the operating point of the unloaded framed circuit lies at the stability margin represented by the red line in Figure 2 defined by (5a). Thus, as expected, it is marginally stable. While we have already shown that PT-symmetric systems are necessarily marginally stable, including the analyzed metasurface-based PT-symmetric system, in practical realization their operating point may easily drift into an unstable region due to the imperfection of the system components. Moreover, to exhibit some of the interesting effects listed in the introduction, PT-symmetric systems usually need to exchange energy with their surroundings. In particular, to exhibit the effects of negative refraction and planar focusing, the metasurface-based PT-symmetric systems proposed in [3] require an incident electromagnetic wave coming from free space, which is then transmitted through the system, and radiated into the forward half-space. Thus, the additional impedances $Z_0$, representing the surrounding free space, are used to load the input and the output port of the system, as shown in Figure 7. Following the proposed stability analysis, it can be shown that such a system is stable providing that:

$$|r| > \frac{\sqrt{2}}{2}. \tag{9}$$

According to [3], the effects of negative refraction and planar focusing occur only for the specific value of the parameter $r = 0.5$. Unfortunately, if $r = 0.5$, the condition (9) is not satisfied, which leads to instability. This instability is what ultimately limits the practical exploitation of the metasurface-based PT-symmetric system proposed in [3]. In [8], the authors reported the occurrence of unstable poles in an acoustic PT-symmetric system based on a circuit model similar to the one in Figure 7. However the mechanism and cause of instability were not further investigated. The instability was avoided by engineering the dispersion of the system elements and reducing the operating bandwidth. The most recent attempts to avoid instability in PT-symmetric systems are given in [16]. Here, the authors proposed an inherently stable, gain-free route to achieve effects similar to those that arise from PT symmetry. The method extends the concept of virtual absorption to implement virtual gain. Since it is fully passive, such a system is not based on the balance of loss and gain, which may limit its performance. Moreover, it may be more sensitive to inevitable parasitic losses ever-present in passive systems. Thus, stability analysis and understanding the stability properties remain essential prerequisites for designing active PT-symmetric systems.

**Figure 7.** Circuit model of a metasurface-based PT-symmetric system that exchanges energy with its environment.

## 5. Conclusions

In this paper we presented a stability analysis of PT-symmetric systems in general, and its verification of the recently introduced model of metasurface-based PT-symmetric systems. The analysis was verified both numerically and experimentally. The investigation lead to the conclusion that any PT-symmetric system is necessarily marginally stable. It was shown that such a marginally stable system may easily become unstable if it exchanges energy with its surroundings. Thus, practical PT-symmetric systems must be designed carefully. While the presented analysis was based on circuit theory and the experimental verification was conducted in a lower radio frequency range, all conclusions can be applied to any PT-symmetric system operating in any frequency range, including optical frequencies.

**Author Contributions:** Conceptualization, J.L. and S.H.; methodology, J.L.; software, J.L.; validation, J.L., I.K. and J.V.; formal analysis, J.L.; investigation, J.L.; resources, J.L. and S.H.; data curation, J.L. and S.H.; writing—original draft preparation, J.L.; writing—review and editing, J.L., S.H. and J.V.; visualization, J.L.; supervision, S.H.; project administration, S.H.; funding acquisition, S.H. All authors have read and agreed to the published version of the manuscript.

**Funding:** This material is based upon the work on the project *"Non-Foster Networks for Tunable and Wideband RF Devices"* supported by EOARD/AFRL, Grant No. FA8655-20-1-7008, and *"Electromagnetic Structures for Emerging Communication Systems"* supported by the HRZZ Grant No. IP 2018-01-9753.

**Conflicts of Interest:** The authors declare no conflict of interest. The funders had no role in the design of the study, in the collection, analyses, or interpretation of data, in the writing of the manuscript, or in the decision to publish the results.

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
