# Peer review of "Stability Constraints on Practical Implementation of Parity-Time-Symmetric Electromagnetic Systems"

_photonics, doi:10.3390/photonics8020056_

Round 1

Reviewer 1 Report

Dear Authors and Editors

This is the review of the article with title "Stability Constraints on Practical Implementation of Parity-time-symmetric Electromagnetic Systems" by Lončar et al.

This work is essentially a stability analysis of a given PT-symmetric system in the RF domain (MHz region). The authors correctly recognize that although the PT-symmetric systems can offer wonderful responses, there might be certain stability issues pertaining their operation.

The work is nicely and clearly written. The figures and the flow are clear and logical. As for the scientific merits, I found the article reasonable and sound. The authors offer essentially an experimental verification to the studied problem, i.e., a cross validation of their results and the results presented in [3] with an additional stability condition extracted. It is certainly a nice work for someone to consider, especially working with such PT-symmetric concepts:

  • One minor critical question is whether and how one might try to adopt their methodology for studying the stability of PT-systems in the photonic or the acoustic domain, such as in gain/loss media studies. Is it possible for someone to perform the same analysis for other domains? If yes, can the authors tell if there might be some limitations or adjustments that someone should perform for studying the stability effects in other domains? Can these results be extended beyond a transmission line/circuit model???
  • Another minor critical question: apart the stability analysis, is there any functionality that the designed circuit can demonstrate, beyond the ones that are shown in example [3]? Did the authors consider that the implemented circuitry might be an excellent playground for other systems where the interplay between the poles and the zeros of the systems can deliver interesting phenomena (e.g. time varying metamaterials, virtual absorption or other similar concepts?)
  • Minor comment: Equation 9 is the extracted stability condition. If I have understood correctly, this condition refers for circuits and PT-systems as the one shown in Figure 7. Can this condition be generalized for other generic "black box" PT-systems?
  • Also, a comment regarding Figure 6 (a): Why the measured envelope has significantly smaller growing ratio? how is this explained??? Also why most of the envelopes do not coincide with the peak values of their corresponding pulses? Is there any practical/technical reason behind this?

In view of the above, I enjoyed reading this simple, yet very instructive work. A proper combination of theory and experimental verification can indeed make the discussion over PT-systems much more interesting and illuminating. The authors could as well consider the above minor comments. In any case, I recommend the acceptance of this work since it is already complete in its current form.

Reviewer 2 Report

The paper is well written. On the other hand, the following suggestions might be considered by the authors as follows:

1) The paper is clear and all the objectives well stated. At the same time, some recent technology developments are missing such as:

_ metamaterials [Inverse-designed metastructures that solve equations, Science 363 (6433), 1333-1338, 2019]

_ plasmonics [Plasmonic Optical and Chiroptical Response of Self-Assembled Au Nanorod Equilateral Trimers, ACS nano, 2019]

_ graphene [Graphene acoustic plasmon resonator for ultrasensitive infrared spectroscopy, Nature Nanotechnology 14, 313–319, 2019]
_ nanoparticles [Modeling, design, and synthesis of gram-scale monodispersed silver nanoparticles using microwave-assisted polyol process for metamaterial applications, Optical Materials 108, 110381, 2020]

It would be beneficial for the reader if authors include such technologies in the introduction section to have a complete picture of the state-of-art.

2) The analytical model to describe the structure behaviour is similar to that one present in [Metamaterial-based wideband electromagnetic wave absorber, Optics express 24 (6), 5763-5772, 2016]
Explain what are the main differences of your model with this one above mentioned

3) Explain in details what are the advantages/disadvantages and similarities/differences of the above-mentioned with yours; and how it can be applied to your structure.

4) To explore the device behaviour, authors can consider the following interesting electromagnetic phenomena: 
_ electric, magnetic and displacements currents [Hybrid bilayer plasmonic metasurface efficiently manipulates visible light, Science Advances, 2(1), 2016]

Include such phenomena in your model and explain how they can affect the device properties.

5) The paper lack in application examples. I would suggest to create a small paragraph by considering applications of your structure and explaining how you can use your device. Please highlight what's new in yours.

6.1) No limitations of the proposed method have been highlighted.

6.2) No future improvements/works have been discussed.

Reviewer 3 Report

[1] Line 120: The definition of delay as “tau = vp/l” seems reversed.

[2] Eqn.(7): The expressions of “omega_0” are inconsistent with those of “f_0”.

[3] Figure 6: The pulse width specified in the caption is inconsistent with that in the plot.

[4] Lie 152: The measurement results in Figure 6 are not the same as the simulated counterparts. Please explain the claim of “absolute agreement”.
